# Phase Transitions in the Interacting Relativistic Boson Systems

Dmitry Anchishkin [1,2,3,†], Volodymyr Gnatovskyy [2,*,†], Denys Zhuravel [1,†], Vladyslav Karpenko [2,†], Igor Mishustin [3,†] and Horst Stoecker [3,4,†]

1   Bogolyubov Institute for Theoretical Physics, 03143 Kyiv, Ukraine; dmytro.kiev@gmail.com (D.A.);
    lpbest@ukr.net (D.Z.)
2   Faculty of Physics, Taras Shevchenko National University of Kyiv, 03127 Kyiv, Ukraine;
    karpych1717@gmail.com
3   Frankfurt Institute for Advanced Studies, 60438 Frankfurt am Main, Germany;
    mishustin@fias.uni-frankfurt.de (I.M.); stoecker@fias.uni-frankfurt.de (H.S.)
4   Johann Wolfgang Goethe University, 60438 Frankfurt am Main, Germany
*   Correspondence: vgnatovskyy@ukr.net
†   These authors contributed equally to this work.

**Abstract:** The thermodynamic properties of the interacting particle–antiparticle boson system at high temperatures and densities were investigated within the framework of scalar and thermodynamic mean-field models. We assume isospin (charge) density conservation in the system. The equations of state and thermodynamic functions are determined after solving the self-consistent equations. We study the relationship between attractive and repulsive forces in the system and the influence of these interactions on the thermodynamic properties of the bosonic system, especially on the development of the Bose–Einstein condensate. It is shown that under "weak" attraction, the boson system has a phase transition of the second order, which occurs every time the dependence of the particle density crosses the critical curve or even touches it. It was found that with a "strong" attractive interaction, the system forms a Bose condensate during a phase transition of the first order, and, despite the finite value of the isospin density, these condensate states are characterized by a zero chemical potential. That is, such condensate states cannot be described by the grand canonical ensemble since the chemical potential is involved in the conditions of condensate formation, so it cannot be a free variable when the system is in the condensate phase.

**Keywords:** relativistic boson system of particles and antiparticles; Bose–Einstein condensation

## 1. Introduction

Knowledge of the phase structure of meson systems in the regime of finite temperatures and isospin densities is crucial for understanding a wide range of phenomena, from nucleus–nucleus collisions to neutron stars, as well as cosmology. This field is an important part of hot and dense hadronic matter research. Meanwhile, the study of meson systems has its own specifics due to the possibility of the Bose–Einstein condensation of bosonic particles. The aim of this paper is to investigate thermodynamic properties of a bosonic many-particle system, specifically the character of the phase transitions during the Bose–Einstein condensation at high densities. The latter condition means that the interaction in the bosonic system plays a sufficient role.

Historically, the problem of the Bose–Einstein condensation in the system of interacting bosons has been studied, starting from the pioneering works of N.N. Bogolyubov [1], where he investigated non-ideal gas of bosons and managed to describe the excitations of the system of interacting bosons in terms of non-interacting quasi-particles. Starting from this approach, the investigation of interacting bosons at the temperatures close to zero yielded the powerful impulse from the mean-field approach. Indeed, if the interactions in the diluted atomic gases are sufficiently weak, it can be argued that the mean field is the condensate wave function itself, as it was argued in Refs. [1–3]. Bogolyubov developed

this idea systematically to study Bose condensation and superfluidity. Then, neglecting the fluctuations altogether, it is possible to derive the equation of motion for the wave function of the mean field, i.e., for the condensate wave function. This is the nonlinear Schrodinger equation or Gross–Pitaevskii equation [4,5]. Afterwards, these approaches were supplemented by the number of fruitful generalizations.

However, these methods and approaches are not suitable for the study of the Bose–Einstein condensation at high densities. That has two main reasons. First, high densities imply that the possible condensate states occupy the region of high temperatures where the density of thermal particles can no longer be treated as a small fluctuation comparing with the density of condensate. Second, as was pointed out by Kerson Huang in his textbook [6], the real conservation law deals with the conserved quantity that is the number of particles minus the number of antiparticles. That is why any study of the Bose–Einstein condensation in the relativistic Bose gas must take antiparticles into account. Firstly, this was discussed in Ref. [7]. Moreover, as was shown in [8] in the case of the "weak" attraction in the system and a conserved charge, the particles only develop the condensate states, but the antiparticles are in the thermal phase for all temperature ranges beginning from zero temperature.

In the present study, we are focused first of all on meson systems. This field is an essential part of investigations of hot and dense hadronic matter, which is a subject of active research [9]. In our study, we name the bosonic particles "pions" conventionally. The preference is made because the charged $\pi$ mesons are the lightest hadrons that couple to the isospin chemical potential. At the same time, the pions are the lightest nuclear boson particles and thus, an account for "temperature creation" of particle–antiparticle pairs is a task for quantum statistics widely exploited in the paper. The problem of the Bose–Einstein condensation of $\pi$ mesons has been studied previously, starting from the pioneering works of A.B. Migdal and coworkers (see [10] for references). Formation of classical pion fields in heavy-ion collisions was discussed in Refs. [11–14], and the systems of pions and K mesons with a finite isospin chemical potential have been considered in more recent studies [15–19]. A scalar model of a bosonic system that develops a Bose–Einstein condensate with conservation of isospin (charge) was first studied in [7,20,21]. Various aspects of free and interacting systems of relativistic bosons are discussed further in Refs. [22–26]. First-principles lattice calculations provide interesting new results concerning dense pion systems [27,28].

The presented study is associated with the approach proposed in Ref. [8], where the boson system was considered when the attraction between particles is "weak". Here, we proceed to investigate the thermodynamic properties of interacting particle–antiparticle meson systems at the conserved isospin density in the framework of the canonical ensemble using the mean-field model (see Appendix A). In this paper, we study also the boson systems where the attractive interaction between particles is "strong". (The rigorous definitions of the "weak" and "strong" attractive interactions will be given further.) We regard a studied self-interacting many-particle system as a toy model that can help us understand the Bose–Einstein condensation and phase transitions over a wide range of temperatures and densities.

The paper is organized as follows. Section 2 shortly describes the thermodynamic properties and condensation in an ideal Boson gas at the particle-number conservation. In Section 3, we introduce a self-interacting scalar mean-field model, which is then used to investigate condensate creation in the bosonic system of particles and antiparticles. An analogous description of the bosonic system of particles and antiparticles, but in the framework of the thermodynamic mean-field model, is given in Section 4. Section 5 compares the results obtained in the former two approaches for describing the bosonic system and the condensate formation at zero total charge. The phase transitions in the particle–antiparticle system with conserved isospin (charge) density are studied in Section 6. Section 7 is a final one, where we compare the description of the Boson systems in the presence of condensate in the framework of the canonical ensemble and the grand canonical ensemble. Conclusions of the present study are given in Section 8.

## 2. Canonical Ensemble: Condensation in Ideal Boson Gas

As a referring point, let us give a reminder about the main properties of the Bose condensation in a single-component ideal gas at conserved particle-number density $n$. This is shown in Figure 1, where two samples of the particle-number density are presented, $n = 0.1, 0.2$ fm$^{-3}$. The red dashed line is the critical curve $n_{\text{lim}}^{(\text{id})}$ that determines the critical temperature $T_{\text{c}}$. The critical curve is the dependence of the particle-number density on the temperature at the maximum value of the chemical potential, which is equal to the particle mass, $\mu = m$ [1]. Thus, the formula that determines the critical curve reads

$$n_{\text{lim}}^{(\text{id})} = g \int \frac{d^3k}{(2\pi)^3} f_{\text{BE}}(E, \mu)\big|_{\mu=m}, \tag{1}$$

where $E = \omega_k = \sqrt{m^2 + k^2}$ and

$$f_{\text{BE}}(E, \mu) = \frac{1}{e^{(E-\mu)/T} - 1}. \tag{2}$$

In Figure 1 and further in the text, the dependence $n(T)$ given in Equation (3) is noted as $n = n_{\text{lim}}^{(\text{id})}(T)$. The solution of Equation (3) with respect to temperature for the given particle density $n$ determines the critical temperature $T_{\text{c}}(n)$.

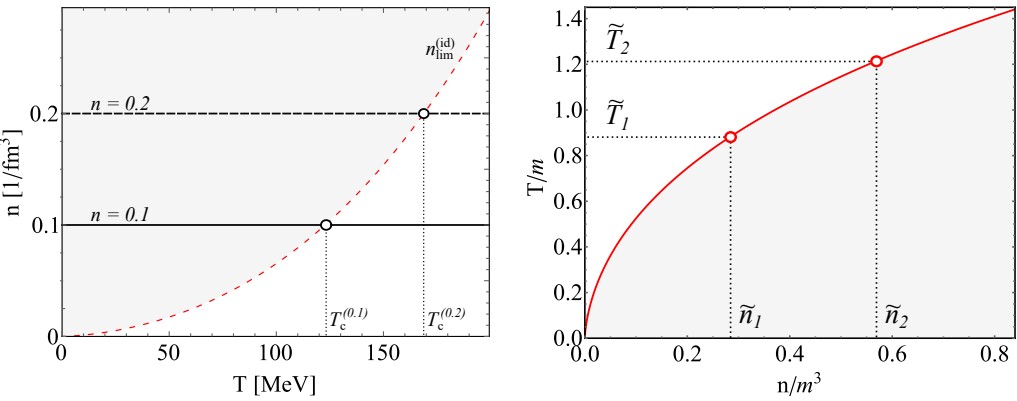

**Figure 1. Left panel:** particle-number density versus temperature in ideal single-component gas. The horizontal lines represent two constant particle density samples, $n = 0.1, 0.2$ fm$^{-3}$, which correspond to critical temperatures $T_{\text{c}}^{(0.1)}$ and $T_{\text{c}}^{(0.2)}$, respectively. Here, the critical curve $n_{\text{lim}}^{(\text{id})}(T)$ is defined in (3). **Right panel:** normalized critical temperature $\tilde{T} = T/m$ vs. normalized particle density $\tilde{n} = n/m^3$ in ideal single-component gas.

In the condensate phase, the generalization of Equation (3) is

$$n = n_{\text{cond}}(T) + g \int \frac{d^3k}{(2\pi)^3} f_{\text{BE}}(E, \mu)\big|_{\mu=m}. \tag{3}$$

The results of calculation of the energy density and heat capacity represented in Figure 2 evidently show that at the crossing point of the particle-number density and the critical curve, the phase transition of the second order occurs. Indeed, there is a finite discontinuity of the derivative of the heat capacity in the $T_{\text{c}}$ points and a smooth behavior of the energy–density dependence in these points, i.e, there is no release of the latent heat.

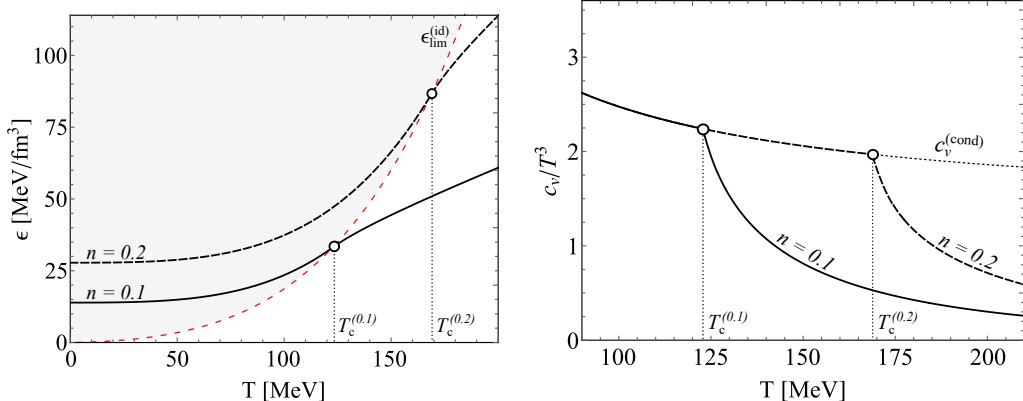

**Figure 2. Left panel:** energy density versus temperature for the same system and conditions as in the left panel. The red dashed line marked as $\varepsilon_{\text{lim}}^{(\text{id})}$ represents the energy density of the states that belong to the critical curve $n_{\text{lim}}^{(\text{id})}$ depicted in the upper panel. **Right panel:** heat capacity normalized to $T^3$ as a function of temperature in the ideal single-component gas where the particle-number density is kept constant.

Let us briefly discuss the results obtained for a single-component ideal gas, where the particle-number density $n$ remains constant. First of all, we fix that when the line $n(T) = \text{const}$ intersects the critical curve $n_{\text{lim}}^{(\text{id})}(T)$, the system undergoes a phase transition of the second order or follows the Ehrenfest classification of the third order. It has long been known, see Ref. [29], that the Bose–Einstein condensation is indeed a third-order phase transition according to the first classification of general types of transitions between phases of matter, introduced by Paul Ehrenfest in 1933 [30,31]. Therefore, the obtained temperature $T_c$ is really the temperature of the phase transition of the second order (according to modern terminology), and the density of condensate $n_{\text{cond}}$ is the order parameter. In what follows, we will show that the same behavior is typical also in the case of interacting two-component systems at conserved charge density.

## 3. Self-Interacting Scalar Field

We start our consideration from the Lagrangian density of the self-interacting real scalar field

$$\mathcal{L}(x) = \frac{1}{2}\left[\partial_\mu \hat{\phi}(x)\partial^\mu \hat{\phi}(x) - m^2\,\hat{\phi}^2(x)\right] + \mathcal{L}_{\text{int}}[\hat{\phi}^2(x)]\,, \tag{4}$$

where $x = (t, \boldsymbol{r})$. We adopt that

$$\hat{\phi}(\boldsymbol{r}) = \phi_{\text{cond}} + \hat{\psi}(\boldsymbol{r})\,, \quad \text{where} \quad \langle\,\hat{\psi}(\boldsymbol{r})\,\rangle = 0\,. \tag{5}$$

Here, we use the famous Bogolyubov's decomposition of the field operator into two contributions [1–3]

$$\hat{\phi}(\boldsymbol{r}) = \frac{1}{\sqrt{V}}a_0 + \frac{1}{\sqrt{V}}\sum_{k\neq 0}a_k e^{ik\cdot r/\hbar}\,. \tag{6}$$

Due to the argument that at $T \to 0$ in a non-perfect Bose gas, the number of particles on the ground state $N_0$ approximately equals to the total number of particles $N$,

$$N_0 = \langle a_0^+ a_0\rangle \approx N\,, \tag{7}$$

one can treat $a_0$ and $a_0^+$ as classical values.

Heisenberg representation:

$$\hat{\phi}(x) = e^{iHt}\hat{\phi}(\boldsymbol{r})e^{-iHt} = \phi_{\text{cond}} + \hat{\psi}(x) \quad \text{with} \quad \langle\hat{\psi}(x)\rangle = 0\,, \tag{8}$$

$$\left[\hat{\psi}(t, \boldsymbol{r}), \frac{\partial \hat{\psi}(t, \boldsymbol{r}')}{\partial t}\right] = \left[\hat{\phi}(t, \boldsymbol{r}), \frac{\partial \hat{\phi}(t, \boldsymbol{r}')}{\partial t}\right] = i\delta^3(\boldsymbol{r} - \boldsymbol{r}'). \tag{9}$$

Hence, the quantum fluctuations of the field $\hat{\psi}(x)$ have the same commutation relation as the complete field $\hat{\phi}(x)$. Expansion over solutions of the Klein–Gordon equation is

$$\hat{\psi}(x) = \int_{|\boldsymbol{p}|\neq 0} \frac{d^3 p}{(2\pi)^3 2\omega_p} \left(a_{\boldsymbol{p}} e^{-ip\cdot x} + a_{\boldsymbol{p}}^+ e^{ip\cdot x}\right)\big|_{p^0=\omega_p}, \tag{10}$$

where

$$[a_k, a_p^+] = (2\pi)^3 2\omega_p \, \delta^3(\boldsymbol{k} - \boldsymbol{p}), \qquad [a_k, a_p] = 0. \tag{11}$$

For the field variance, we obtain the following decomposition:

$$\left\langle \hat{\phi}^2(x) \right\rangle = \left\langle \phi_{\text{cond}}^2 + 2\phi_{\text{cond}}\hat{\psi} + \hat{\psi}^2 \right\rangle = \phi_{\text{cond}}^2 + \left\langle \hat{\psi}^2 \right\rangle. \tag{12}$$

We see that the field variance is decomposed also on classical and quantum pieces.

### 3.1. The Effective Lagrangian in the Mean-Field Approximation

We are going to consider the Bose–Einstein condensation of the scalar field (for details see Ref. [32]),

$$\mathcal{L}(x) = \frac{1}{2}\left[\partial_\mu \hat{\phi}(x)\, \partial^\mu \hat{\phi}(x) - m^2\, \hat{\sigma}(x)\right] + \mathcal{L}_{\text{int}}(\hat{\sigma}), \tag{13}$$

where we introduced notation

$$\hat{\sigma}(x) = \hat{\phi}^2(x). \tag{14}$$

We use the quantum statistical averaging of the operator $\hat{A}$:

$$\left\langle \hat{A} \right\rangle = \frac{1}{Z}\text{Tr}\left[e^{-\beta\left(\hat{H}-\mu\hat{N}\right)}\hat{A}\right], \qquad Z = \text{Tr}\left[e^{-\beta\left(\hat{H}-\mu\hat{N}\right)}\right]. \tag{15}$$

Next, we introduce the mean value $\sigma$ of the operator $\hat{\sigma}$

$$\sigma = \langle \hat{\sigma} \rangle, \qquad \delta\hat{\sigma} = \hat{\sigma} - \sigma. \tag{16}$$

Here, $\delta\hat{\sigma}$ is the deviation of the operator $\hat{\sigma}$ from its mean value. One can expand the Lagrangian (13) as the function on the variable $\hat{\sigma}$ around the point $\sigma$:

$$\mathcal{L}_{\text{int}}(\hat{\sigma}) \simeq \mathcal{L}_{\text{int}}(\sigma) + \delta\hat{\sigma}\, \mathcal{L}'_{\text{int}}(\sigma) = \mathcal{L}_{\text{int}}(\sigma) + \hat{\sigma}\, \mathcal{L}'_{\text{int}}(\sigma) - \sigma\, \mathcal{L}'_{\text{int}}(\sigma), \tag{17}$$

where prime means the derivative with respect to $\sigma$. We come to the effective Lagrangian in the mean-field approximation

$$\mathcal{L}(x) \simeq \frac{1}{2}\left[\partial_\mu \hat{\phi}(x)\, \partial^\mu \hat{\phi}(x) - M^2(\sigma)\, \hat{\phi}^2(x)\right] + P_{\text{ex}}(\sigma), \tag{18}$$

where we introduced the following notations

$$P_{\text{ex}}(\sigma) \equiv \mathcal{L}_{\text{int}}(\sigma) - \sigma\frac{\partial \mathcal{L}_{\text{int}}(\sigma)}{\partial \sigma}, \qquad \hat{M}^2(\sigma) = m^2 + 2\,U(\sigma), \tag{19}$$

with

$$U(\sigma) \equiv -\frac{\partial \mathcal{L}_{\text{int}}(\sigma)}{\partial \sigma}. \tag{20}$$

The differential relation between the excess pressure $P_{\text{ex}}(\sigma)$ and the mean field $U(\sigma)$ follows from this definition

$$\sigma\frac{\partial U(\sigma)}{\partial \sigma} = \frac{\partial P_{\text{ex}}(\sigma)}{\partial \sigma}. \tag{21}$$

### 3.2. Hamiltonian Density in the Mean-Field Approximation

Momentum operator $\hat{\pi}$ satisfies the equal-time commutation relations

$$\hat{\pi}(x) = \partial_t \hat{\phi}(x), \qquad \left[\hat{\phi}(t, \boldsymbol{r}), \hat{\pi}(t, \boldsymbol{r}')\right] = i\delta^3(\boldsymbol{r} - \boldsymbol{r}'). \tag{22}$$

The Hamiltonian density $\hat{\mathcal{H}} = \hat{\pi} \, \partial_t \hat{\phi} - \mathcal{L}$ reads

$$\hat{\mathcal{H}} \simeq \frac{1}{2}\left[\hat{\pi}^2(x) + \boldsymbol{\nabla}\hat{\phi}(x) \cdot \boldsymbol{\nabla}\hat{\phi}(x) + M^2(\sigma)\hat{\phi}^2(x)\right] - P_{\text{ex}}(\sigma). \tag{23}$$

Using solutions of the Klein–Gordon equation,

$$\partial^\mu \partial_\mu \, \hat{\phi} + M^2(\sigma) \, \hat{\phi} = 0, \tag{24}$$

one can represent the scalar field $\hat{\phi}(x)$ as

$$\hat{\phi}(x) = g \int \frac{d^3k}{(2\pi)^3 \sqrt{2\omega_k}} \left[a_{\boldsymbol{k}} e^{-ik \cdot x} + a_{\boldsymbol{k}}^+ e^{ik \cdot x}\right], \tag{25}$$

where $k^0 = \omega_{\boldsymbol{k}} = \sqrt{\boldsymbol{k}^2 + M^2(\sigma)}$ and the operators of creation and annihilation satisfy the standard commutation relations

$$\left[a_{\boldsymbol{k}}, a_{\boldsymbol{k}'}^+\right] = (2\pi)^3 \delta(\boldsymbol{k} - \boldsymbol{k}'), \qquad \left[a_{\boldsymbol{k}}, a_{\boldsymbol{k}'}\right] = \left[a_{\boldsymbol{k}}^+, a_{\boldsymbol{k}'}^+\right] = 0. \tag{26}$$

As a first step, we consider a boson system at zero isospin (charge) density $n_I = 0$, i.e., the numbers of particles and antiparticles are equal. In this case, the Hamiltonian in the mean-field (MF) approximation reads

$$\hat{H} = \int d^3x \, \hat{\mathcal{H}} = V\left[g \int \frac{d^3k}{(2\pi)^3} \, \omega_{\boldsymbol{k}} \, a_{\boldsymbol{k}}^+ a_{\boldsymbol{k}} - P_{\text{ex}}(\sigma)\right]. \tag{27}$$

In the MF approximation, the equilibrium momentum distribution coincides with that of an ideal gas of bosons with the effective mass $M(\sigma)$

$$n_{\boldsymbol{k}}(\sigma) \equiv \langle a_{\boldsymbol{k}}^+ a_{\boldsymbol{k}} \rangle = (e^{\beta \omega_{\boldsymbol{k}}} - 1)^{-1}, \qquad \beta = 1/T, \quad k_{\text{B}} = 1, \quad \mu_I = 0, \tag{28}$$

where $\omega_{\boldsymbol{k}} = \sqrt{M^2(\sigma) + \boldsymbol{k}^2}$ with $M^2(\sigma) = m^2 + 2U(\sigma)$.

The thermodynamical description of the system is obtained by means of solution of self-consistent equations for the thermal phase and condensate phase with respect to the scalar density $\sigma = \langle \hat{\phi}^2 \rangle$ [2]. In the thermal phase, this equation reads

$$\sigma = g \int \frac{d^3k}{(2\pi)^3} \frac{n_{\boldsymbol{k}}(\sigma)}{\omega_{\boldsymbol{k}}}. \tag{29}$$

In the condensate phase, one should take into account the necessary condition for condensate creation $M^2(\sigma) = 0$ and include into the equation the density of the scalar condensate, then the equation becomes

$$\sigma = \sigma_{\text{cond}} + g \int \frac{d^3k}{(2\pi)^3} \frac{n_{\boldsymbol{k}}(\sigma)}{\omega_{\boldsymbol{k}}}\bigg|_{M^2(\sigma)=0}, \tag{30}$$

where, in the case of $\mu_I = 0$ (or $n_I = 0$), we are left with one canonical variable $T$. The last equation corresponds to the relation

$$\left\langle \hat{\phi}^2 \right\rangle = \phi_{\text{cond}}^2 + \left\langle \hat{\psi}^2 \right\rangle, \tag{31}$$

which we obtained as a result of the decomposition of the field operator (5) and specific features of the quantum fluctuations, see Equation (12).

Other thermodynamic quantities that characterize the quasi-particle boson system can be obtained in a regular way in the framework of the quantum statistics. The pressure reads

$$p = p_{\text{kin}}(T, \sigma) + P_{\text{ex}}(\sigma), \tag{32}$$

where the kinetic pressure in the thermal phase is

$$p_{\text{kin}}(T, \sigma) = \frac{g}{3} \int \frac{d^3 k}{(2\pi)^3} \frac{k^2}{\omega_k} n_k(\sigma), \tag{33}$$

whereas the kinetic pressure in the condensate phase reads

$$p_{\text{kin}}(T, \sigma) = \frac{g}{3} \int \frac{d^3 k}{(2\pi)^3} \frac{k^2}{\omega_k} n_k(\sigma) \Big|_{M^2(\sigma) = 0}. \tag{34}$$

The energy density and entropy density $s = (\varepsilon + p)/T$ in the thermal phase read

$$\varepsilon = g \int \frac{d^3 k}{(2\pi)^3} \omega_k n_k(\sigma) - P_{\text{ex}}(\sigma), \tag{35}$$

$$s = \frac{g}{T} \int \frac{d^3 k}{(2\pi)^3} \left( \omega_k + \frac{k^2}{3\omega_k} \right) n_k(\sigma). \tag{36}$$

The energy density and entropy density in the condensate phase read

$$\varepsilon = \varepsilon_{\text{cond}} + g \int \frac{d^3 k}{(2\pi)^3} \omega_k n_k(\sigma) \Big|_{M^2(\sigma) = 0} - P_{\text{ex}}(\sigma), \tag{37}$$

$$s = s_{\text{cond}} + \frac{g}{T} \int \frac{d^3 k}{(2\pi)^3} \left( \omega_k + \frac{k^2}{3\omega_k} \right) n_k(\sigma) \Big|_{M^2(\sigma) = 0}. \tag{38}$$

*3.3. Bosonic System with $\varphi^4 + \varphi^6$ Self-Interaction*

For specific numerical calculations, we adopt the following parametrization of the interaction part of the Lagrangian

$$\mathcal{L}_{\text{int}}\left(\hat{\phi}^2(x)\right) = \frac{a}{4} \hat{\phi}^4(x) - \frac{b}{6} \hat{\phi}^6(x). \tag{39}$$

Then, the mean field and the excess pressure are

$$U(\sigma) = -\frac{1}{2} a \sigma + \frac{1}{2} b \sigma^2, \qquad P_{\text{ex}}(\sigma) = -\frac{a}{4} \sigma^2 + \frac{b}{3} \sigma^3, \tag{40}$$

where $\sigma = \langle \hat{\phi}^2 \rangle$. This means that attraction and repulsion between particles in the form of a mean field are simultaneously present in the system of bosons. The distribution function $n_k = \left[ \exp\left(\sqrt{k^2 + M^2}/T\right) - 1 \right]^{-1}$ makes sense when the argument is positive, i.e.,

$$M^2(\sigma) = m^2 + 2U(\sigma) = m^2 - a\sigma + b\sigma^2 \geqslant 0. \tag{41}$$

The limiting case in this relation is the condition for the occurrence of scalar condensate:

$$M^2(\sigma) = m^2 - a\sigma + b\sigma^2 = 0. \tag{42}$$

Solutions of this equation are

$$\sigma_{1,2} = \frac{m}{\sqrt{b}} \left( \kappa \mp \sqrt{\kappa^2 - 1} \right), \tag{43}$$

where we introduce the dimensionless parameter $\kappa$:

$$\kappa = \frac{a}{2m\sqrt{b}}, \quad \rightarrow \quad a = \kappa\, a_c, \quad a_c = 2m\sqrt{b}. \tag{44}$$

Thus, we conclude that when $\kappa \leq 1$, the quasi-particle effective mass becomes imaginary ($M^2 < 0$), and the system becomes unstable. The stability is restored by the formation of the Bose condensate.

## 4. An Interacting Boson System within the Thermodynamic Mean-Field Model

We are going to compare a description of the boson system at high densities in the field-theoretical and in the quantum-statistical approaches. The consideration of the latter one starts from separation of the free energy $F(T, N, V)$ into free and interaction parts as

$$F(T, N, V) = F_0 + F_{\text{int}}. \tag{45}$$

Or, for the free energy density as $\Phi(T, n) = \Phi_0 + \Phi_{\text{int}}$, where $\Phi(T, n) = F(T, N, V)/V$. Then, we introduce the following important notations (for details, see [33])

$$U(n, T) = \left[ \frac{\partial \Phi_{\text{int}}(n, T)}{\partial n} \right]_T, \tag{46}$$

$$P_{\text{ex}}(n, T) = n \left[ \frac{\partial \Phi_{\text{int}}(n, T)}{\partial n} \right]_T - \Phi_{\text{int}}(n, T). \tag{47}$$

These quantities are related to one another by the differential equality

$$n \frac{\partial U(n, T)}{\partial n} = \frac{\partial P_{\text{ex}}(n, T)}{\partial n}. \tag{48}$$

In these notations, the pressure in the system can be written as

$$p(T, \mu_I) = \frac{g}{3} \int \frac{d^3 k}{(2\pi)^3} \frac{k^2}{\sqrt{m^2 + k^2}} f_{\text{BE}} \left( E_k(n), \mu_I \right) + P_{\text{ex}}(n), \tag{49}$$

where $g$ is the degeneracy factor, $E_k(n) = \sqrt{m^2 + k^2} + U(n)$ is the effective single-particle energy, $\mu_I$ is the isospin chemical potential and $f_{\text{BE}}$ is the Bose–Einstein distribution function

$$f_{\text{BE}}(E, \mu) = \left\{ \exp\left[ \frac{E - \mu}{T} \right] - 1 \right\}^{-1}. \tag{50}$$

In the particle–antiparticle system, the Euler relation is $\varepsilon + p = Ts + \mu_I n_I$, where $n_I$ is the isospin (charge) density. Let us first consider the case of zero charge density, i.e., $n_I = 0$, that corresponds to $\mu_I = 0$ in the grand canonical ensemble.

The mean-field model implies that the thermodynamic description of the system is obtained via a self-consistent approach. In our case, this is achieved by a self-consistent equation for the total particle density $n$, which should be solved separately in the thermal and condensate phases. In the thermal phase, this equation has a structure $n = n_{\text{th}}(T, n)$, and it should be solved with respect to the total particle density $n$ for every fixed value of $T$,

$$n = g \int \frac{d^3 k}{(2\pi)^3} f_{\text{BE}}(E_k(n)), \tag{51}$$

where $f_{\mathrm{BE}}(E) = [\exp{(E/T)} - 1]^{-1}$. The solution of Equation (51) in the thermal phase results in the explicit dependence $n = n(T)$, which in general differs from the ideal gas dependence, $n_0(T)$. Knowledge of the dependence $n(T)$ gives a possibility to obtain equation of state through a direct calculation of other thermodynamic quantities such as pressure, energy density, entropy density, etc.

In the condensate phase, one should take into account the condensation condition at $\mu_I = 0$, $U(n) + m = 0$ that leads to specific "critical" density $n_{\mathrm{c}}$, which is a real root of this equation. The solution of this equation has the following structure: $n_{\mathrm{c}} = n_{\mathrm{cond}}(T) + n_{\mathrm{th}}(T)$, where the density of the condensate component $n_{\mathrm{cond}}$ appears as a new degree of freedom. Thus, in the condensate phase, the self-consistent equation for $n_{\mathrm{cond}}$ reads

$$n_{\mathrm{c}} = n_{\mathrm{cond}} + g \int \frac{d^3k}{(2\pi)^3} f_{\mathrm{BE}}(E_{\mathrm{kin}}), \tag{52}$$

where $E_{\mathrm{kin}} = \sqrt{m^2 + k^2} - m$.

### 4.1. Parametrization of the Interaction

To be closer to the field-theoretical approach, we use the following correspondence between the scalar density $\langle \phi^2 \rangle$ and particle number density $n$, which simply coincide with one another in the non-relativistic limit. Then, using the correspondence $\varphi^4 \rightarrow n^2$ and $\varphi^6 \rightarrow n^3$, we write the excess pressure and the corresponding mean field (see the differential relation (48)) as

$$P_{\mathrm{ex}}(n) = -\frac{1}{2} A n^2 + \frac{2}{3} B n^3, \qquad \rightarrow \qquad U(n) = -A n + B n^2, \tag{53}$$

where the positive parameter $A$ is responsible for attraction between particles and the positive parameter $B$ for repulsion between particles in a boson system (for details see [10]). The parameter $A$ will be varied, whereas the parameter $B$, associated with a hard-core repulsion, will be kept constant. It is advisable to parameterize $A$ in the following way: let us use solutions of equation $U(n) + m = 0$, which determine the condition for a condensate creation (a similar algorithm was adopted in Refs. [10,34]). For the given mean field (53), there are two roots of this equation

$$n_1 = \sqrt{\frac{m}{B}} \left( \kappa - \sqrt{\kappa^2 - 1} \right), \qquad n_2 = \sqrt{\frac{m}{B}} \left( \kappa + \sqrt{\kappa^2 - 1} \right), \tag{54}$$

where we introduce the dimensionless parameter $\kappa$:

$$\kappa \equiv \frac{A}{2\sqrt{m B}}. \tag{55}$$

Then, one can parameterize the attraction coefficient as $A = \kappa A_{\mathrm{c}}$ with $A_{\mathrm{c}} = 2\sqrt{mB}$. As it is seen below, the parameter $\kappa$ is the scale parameter that determines the phase structure of the system. We consider two intervals of the parameter $\kappa$: (1) a "weak" attraction that corresponds to $\kappa < 1$, i.e., $n_{1,2}$ are not the real roots, and (2) a "strong" attraction that corresponds to $\kappa > 1$, i.e., $n_{1,2}$ are the real roots. The critical value $A_{\mathrm{c}}$ is obtained when both roots coincide, i.e., when $\kappa = 1$, then $A = A_{\mathrm{c}} = 2\sqrt{mB}$.

## 5. Condensation of Interacting Bosons at Finite Temperatures

In this section, we compare the numerical results obtained within two approaches, the field-theoretical approach, which is based on the scalar mean-field (SMF) model, and the quantum-statistical approach, which is based on the thermodynamic mean-field (TMF) model. Our purpose is to study an influence of the attraction and repulsion between particles on the thermodynamic properties of a Boson system, especially in the presence of the condensate. In both cases, we will keep constant the repulsive term while varying the at-

tractive interaction by means of the parameter $\kappa$. We present the solutions of self-consistent equations for different values of the attraction coefficient $a$ in the SMF model while fixing the repulsion coefficient as $b = 25\, m_\pi^{-2}$. The same is done for the TMF model, where we vary the attraction coefficient $A$ while fixing the repulsion coefficient as $B/m_\pi = 10v_0^2$. It is necessary to note that these variations of the attraction coefficients are done in the same way in both approaches by means of the dimensionless parameter $\kappa$: in the SMF model as $a = \kappa a_c$, where $a_c = 2m\sqrt{b}$, and in the TMF model as $A = \kappa A_c$, where $A_c = 2\sqrt{mB}$. We name the boson particles "pions" and take their mass as $m = m_\pi = 139$ MeV for the degeneracy factor $g = 3$. In the SMF model, the critical curve is obtained when $M^2 = m^2 + 2U(n) = 0$, and it reads

$$\sigma_{\lim} = g \int \frac{d^3k}{(2\pi)^3} \left( e^{k/T} - 1 \right)^{-1} = \frac{g}{12} T^2 . \tag{56}$$

In the case of the TMF model, there is a similar condition for determination of the critical curve, $m + U(n) = 0$, that looks like a presence of the effective chemical potential $\mu = m$. Therefore, the critical curve in the case of the TMF model reads

$$n_{\lim} = g \int \frac{d^3k}{(2\pi)^3} \left\{ \exp\left[ \frac{\sqrt{m^2 + k^2} - m}{T} \right] - 1 \right\}^{-1} . \tag{57}$$

The numerical calculations of the particle density vs, temperature for the SMF model and TMF model are presented in Figure 3 in the left and right panels, respectively. The calculations are done for different values of the attraction coefficients $a$ and $A$, which are parameterized by parameter $\kappa$ in both models. We name $\kappa < 1$ as the "weak" attraction and $\kappa > 1$ as the "strong" attraction. It is seen that for "weak" attraction, the scalar densities and the particle number densities are in the thermal phase. At $\kappa = \kappa_c = 1$, the density curves have one common point with the critical (red dashed line). The critical curve $\sigma_{\lim}(T)$ is depicted in Figure 3 in the left panel as a red dotted-dashed line, and the critical curve $n_{\lim}(T)$ is depicted in Figure 3 in the right panel as a red dashed line. In both approaches, at "strong" attraction, $\kappa > 1$, there is a first-order phase transition at $T = T_c$ with creation of the condensate. This is a result of competition of pressure corresponding to two different solutions of the self-consistent equation in the thermal and in the condensate phases.

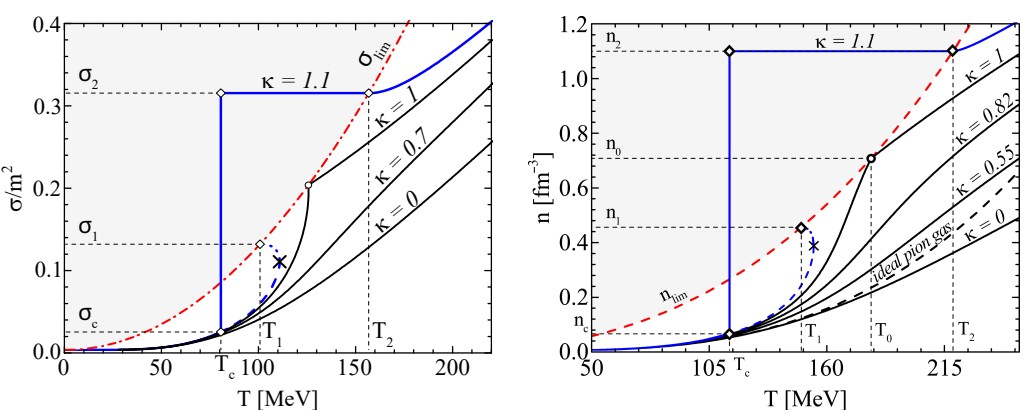

**Figure 3. Left panel:** scalar density vs. temperature, $b = 25\, m_\pi^{-2}$, $a = \kappa a_c$, $a_c = 2m\sqrt{b}$. **Right panel:** particle-number density vs. temperature, $B = 10m_\pi v_0^2$, $A = \kappa A_c$, $A_c = 2\sqrt{mB}$. In both panels, the shaded area indicates the states of the Bose–Einstein condensate. Crosses on both panels separate metastable and non-physical states.

In the SMF model, we solve Equation (30) to obtain the scalar density $\sigma = \sigma_{\text{therm}}(T)$ in the thermal (liquid–gas) phase and a corresponding pressure $p_{\text{lg}}(T)$. On the other hand, Equation (30) for the condensate (mix) phase [3] is characterized by two constant solutions $\sigma = \sigma_1$ and $\sigma = \sigma_2$, see Figure 3, the left panel. Then, we compare the pressure dependencies $p_{\text{mix}}^{(1)}(T)$ and $p_{\text{mix}}^{(2)}(T)$ corresponding to $\sigma_1$ and $\sigma_2$, respectively, with one

another and with $p_{\text{lg}}(T)$. The result of this comparison is depicted in Figure 4 in left panel as the solid blue line. It is seen that at $T = T_{\text{c}}$, the pressure $p_{\text{mix}}^{(2)}(T)$ becomes the largest, which determines the phase transition of the first order with creation of the scalar condensate (for details, see [32]).

The same analysis is made also for the TMF model. We solve Equation (51) to obtain the dependence of the particle density $n = n_{\text{therm}}(T)$ in the thermal (liquid–gas) phase and a corresponding pressure $p_{\text{lg}}(T)$. Equation (52) is characterized by two constant real roots $n_{\text{c1}}$ and $n_{\text{c2}}$ and two corresponding pressures. The result of this comparison is depicted in Figure 4 in the right panel as the solid blue line.

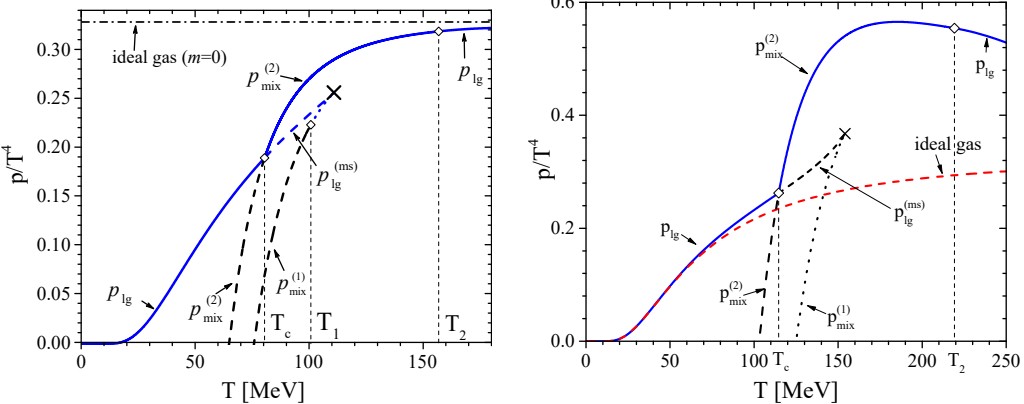

**Figure 4. In both panels:** Pressure vs. temperature for the supercritical attraction, $\kappa = 1.1$. The solid blue line that consists of two segments, $p_{\text{lg}}$ and $p_{\text{mix}}^{(2)}$, is the final equation of state, $T_{\text{c}}$ is the critical temperature that indicates the phase transition of the first order. Crosses on both panels separate metastable and non-physical states. **Left panel:** The scalar mean-field model. The pressure $p_{\text{mix}}^{(1)}$ corresponds to the scalar density $\sigma_1$. **Right panel:** The thermodynamic mean-field model. The pressure $p_{\text{mix}}^{(1)}$ corresponds to the particle-number density $n_1$.

It is seen from the comparison of results depicted in the two panels in Figure 3, and in the two panels in Figure 4, that the two models show a very similar behavior. That is why in what follows, we consider only the TMF model, assuming that it gives a true thermodynamic description of the bosonic system at high densities.

## 6. Particle–Antiparticle System with Conserved Isospin (Charge) Density

### 6.1. Derivation of Basic Equations

Let us consider a homogeneous system with conserved charge (isospin). The description of such a system can be done within the canonical ensemble with the canonical variables $(T, n_I)$. Here, $n_I = n^{(-)} - n^{(+)}$ is the difference between the densities of $\pi^-$ and $\pi^+$ mesons. Note, now we use the thermodynamic mean-field (TMF) model for many-component boson systems, see Appendix A. As a first step, we consider the "weak" attraction between particles, i.e., $\kappa \leq 1$. In this case, there are two pairs of self-consistent equations. The first set of equations describes the system when both components, i.e., both the $\pi^-$ and $\pi^+$ mesons, are in the thermal phase (high temperatures). The second set of equations describes the system at low temperatures, when the $\pi^-$ mesons are in the condensate phase but the $\pi^+$ mesons are in the thermal (kinetic) phase (see details in Ref. [8]). At high temperatures, the set of equations reads

$$n = \int \frac{d^3k}{(2\pi)^3} \left[ f_{\text{BE}}\big(E(k,n), \mu_I\big) + f_{\text{BE}}\big(E(k,n), -\mu_I\big) \right], \tag{58}$$

$$n_I = \int \frac{d^3k}{(2\pi)^3} \left[ f_{\text{BE}}\big(E(k,n), \mu_I\big) - f_{\text{BE}}\big(E(k,n), -\mu_I\big) \right], \tag{59}$$

where the Bose–Einstein distribution function $f_{\mathrm{BE}}(E, \mu)$ is defined in (2) and $E(k, n) = \sqrt{m^2 + k^2} + U(n)$. These equations should be solved with respect to the particle density $n$ and chemical potential $\mu_I$. We use the same parameterization as in the case of zero charge density (see Section 4.1), it depends on the total particle-number density $n$: $P_{\mathrm{ex}}(n) = -(1/2)An^2 + (2/3)Bn^3 \rightarrow U(n) = -An + Bn^2$. Actually, this parameterization of the interaction is in analogy to the field-theoretical approach with a correspondence $\langle \varphi^+ \varphi \rangle \leftrightarrow n$, then, in the same manner, one can write $\varphi^4 \rightarrow n^2$ and $\varphi^6 \rightarrow n^3$.

One of the main goals of our research is to investigate the influence of attraction and repulsion between particles on the thermodynamic properties of the bosonic system, especially in the presence of a condensate. In this study, we fix the repulsive interaction in the system while changing the attraction between particles. As in the case of zero isospin density [10], we use the same parameterization of the attraction coefficient $A$ using the solutions (54) of equation $U(n) + m = 0$. Then, in the same manner, we introduce dimensionless coefficient $\kappa \equiv A/(2\sqrt{mB})$ that parameterizes the parameter $A$ as $A = \kappa A_c$ with $A_c = 2\sqrt{mB}$. Below, we use parameter $\kappa$ to vary attraction between particles.

If one of the components of the particle–antiparticle system is in the condensate phase (low temperatures) [4], then self-consistent equations that determine the thermodynamic structure of the system read

$$n = n_{\mathrm{cond}}^{(-)}(T) + n_{\mathrm{lim}}(T) + \int \frac{d^3k}{(2\pi)^3} f_{\mathrm{BE}}(E(k, n), -\mu_I)\Big|_{\mu_I = m + U(n)}, \qquad (60)$$

$$n_I = n_{\mathrm{cond}}^{(-)}(T) + n_{\mathrm{lim}}(T) - \int \frac{d^3k}{(2\pi)^3} f_{\mathrm{BE}}(E(k, n), -\mu_I)\Big|_{\mu_I = m + U(n)}, \qquad (61)$$

where we assume that the condensed state of $\pi^-$ mesons develops under the (necessary) condition

$$m + U(n) - \mu_I = 0. \qquad (62)$$

We use notation

$$n_{\mathrm{lim}}(T) = \int \frac{d^3k}{(2\pi)^3} f_{\mathrm{BE}}(\omega_k, \mu_I)\Big|_{\mu_I = m} \qquad (63)$$

for a density of the thermal particles at the onset of condensation (the critical curve).

*6.2. Numerical Results: Second-Order Phase Transitions Generated by the Particles That Carry Dominant Charge*

The solutions of Equations (58)–(61) are depicted in Figure 5 as the dependence of particle-number densities of $\pi^-$ mesons (left panel) and $\pi^+$ mesons (right panel) at fixed isospin density $n_I = 0.1$ fm$^{-3}$ and a set of attraction parameters $\kappa = 0, 0.6, 0.85, 0.96, 1$. The red dashed lines in both panels are the critical curves $n_{\mathrm{lim}}$, which reflect the maximal density of thermal $\pi^-$ pions (left panel) or $\pi^+$ pions (right panel). The dashed area indicates the phase with the condensed particles. The open stars in the left panel indicate the Bose condensation as a phase transition of the second order in the $\pi^-$ component, where $T_c^{(-)}$ is the temperature of the Bose condensation of $\pi^-$ mesons. The "dark" star in the right panel indicates a virtual-like second-order phase transition created by the $\pi^+$ meson subsystem at the attraction parameter $\kappa = 1$. Each "star" on the graphs corresponds to a second-order phase transition. Roughly speaking, each intersection of the particle density curve with the critical curve corresponds to a phase transition of the second order.

It turns out that the phase structure of $\pi^-$ mesons (the particles with dominant charge) can be grouped into two types: (a) the curve $n = n^{(-)}(T)$ has one cross with the critical curve $n_{\mathrm{lim}}(T)$, and (b) the curve $n = n^{(-)}(T)$ has three crosses with the critical curve. The regular behavior or the type (a) occurs when the parameter $\kappa$ belongs to the low interval $0 \leq \kappa < \kappa_s$, where $\kappa_s \approx 0.93$. In this case, $\pi^-$ mesons for $T < T_{c1}^{(-)}$ are in the condensate phase, and in the temperature interval $T > T_{c1}^{(-)}$ they are in the thermal phase, see Figure 5, the left panel.

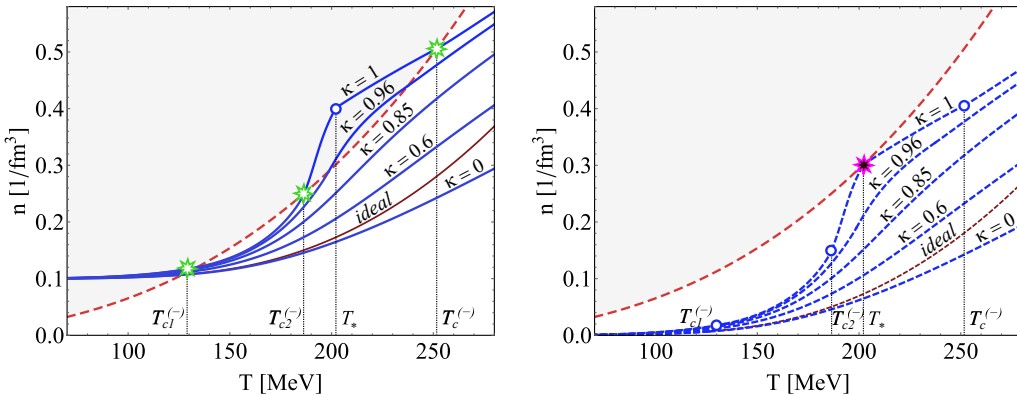

**Figure 5. Left panel:** The particle-number densities $n^{(-)}$ of $\pi^-$ mesons versus temperature for the system of interacting $\pi^+$ - $\pi^-$ mesons at fixed isospin density $n_I = 0.1$ fm$^{-3}$ and the set of "weak" attraction parameters $\kappa = 0$, 0.6, 0.85, 0.96, 1. The red dashed curve $n_{\text{lim}}$ reflects the maximal density of thermal $\pi^-$ mesons (or $\pi^+$ mesons) in the ideal $\pi^+$ - $\pi^-$ gas. The dashed area indicates the phase with the condensed particles. The open stars show the onset of phase transition of the second order of the $\pi^-$ mesons. **Right panel:** The particle-number densities $n^{(+)}$ of $\pi^+$ mesons versus temperature at the same set of parameters as in the left panel. The "dark" star corresponding to the $T_*$ temperature indicates a virtual second-order phase transition of the $\pi^+$ component without condensate formation.

Therefore, for the $\kappa$ of type (a), the temperature of the phase transition $T_c$ in the whole system is determined as $T_c = T_{c1}^{(-)}$, or it is a regular phase transition of the second order. Indeed, in Figure 6, in the left panel, one can clearly see a finite discontinuity of the derivative of heat capacity (left panel) and the absence of the latent heat (right panel) at the temperature $T_c$.

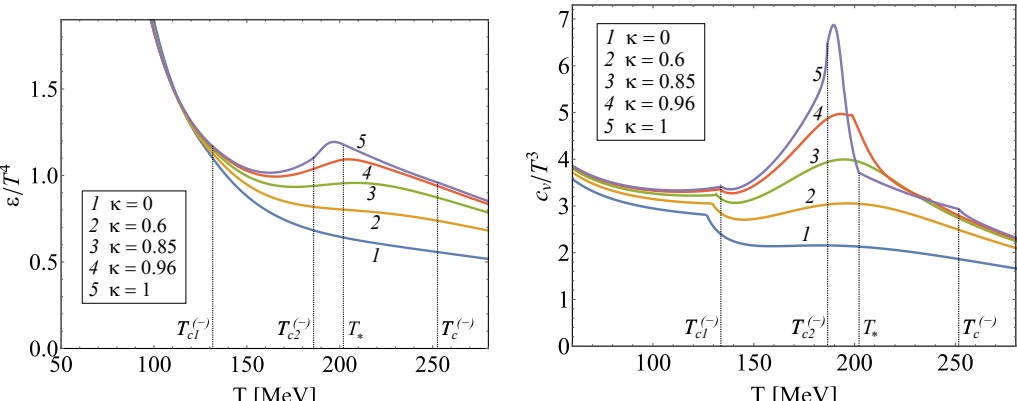

**Figure 6. Left panel:** Energy density versus temperature in the interacting particle–antiparticle system of pions at $\kappa = 0$, 0.6, 0.85, 0.96, 1. The isospin (charge) density is kept constant, $n_I = 0.1$ fm$^{-3}$. The points of the phase transition of the second order are indicated by the corresponding temperatures $T_{c1}^{(-)}$, $T_{c2}^{(-)}$, $T_*$, $T_c^{(-)}$. **Right panel:** heat capacity as a function of temperature for the same system and conditions as in the left panel.

In fact, in case (a), the dependence $n^{(-)}(T)$, which reflects the behavior of the density of $\pi^-$ mesons (Figure 5, left panel), looks very similar to the behavior at a constant density of particles in a single-component system, at least in the condensate phase, that is, for temperatures $0 \leq T \leq T_c$, see Figure 1 in Section 2. On the other hand, the dependence $n^{(+)}(T)$ (Figure 5, right panel) that reflects behavior of the $\pi^+$ particle density looks very similar to the particle-density dependence at $n_I = 0$ and $\kappa < 1$, shown in Figure 3 in the right panel. Both of these features can be explained by the similar initial conditions at $T = 0$ and a slow creation of the thermal pion pairs at low temperatures.

When the attraction parameter $\kappa$ increases and becomes type (b), i.e., $\kappa_s < \kappa \leq 1$, the phase structure of the charge-dominant component ($\pi^-$ mesons) is more complex. In this case, the curve $n^{(-)}(T)$ consistently crosses the critical curve $n_{\lim}(T)$ three times at temperatures $T_{c1}^{(-)} < T_{c2}^{(-)} < T_c$, see Figure 5, the left panel. The obvious explanation of this phenomenon is due to the charge conservation. Indeed, for sufficiently high values of $\kappa$, say $\kappa > \kappa_s$, the $\pi^+$ density approaches the critical curve (see Figure 5, right panel) and simply "squeezes out" to the other side of the critical curve the $\pi^-$ density since its values must be higher by $n_I$ than $\pi^+$ density. That is, the states of the $\pi^-$ mesons again "pass" into the condensate phase. As can be seen in Figure 6, each intersection of the curve $n^{(-)}(T)$ with the critical curve $n_{\lim}(T)$ corresponds to a phase transition of the second order. Indeed, in Figure 6, in the right panel, one can see a finite discontinuity of the derivative of heat capacity at temperatures $T_{c1}^{(-)}$, $T_{c2}^{(-)}$, $T_*$ and $T_c$. At the same time, in the left panel in Figure 6, we see no jumps corresponding to the latent heat at these temperatures. Therefore, we can conclude that due to the conservation of charge, along with the regular phase transition of the second order, multiple "weak" phase transitions can also occur in a particle–antiparticle system.

At the same time, the antiparticle component of the system or $\pi^+$ mesons are in the thermal phase for the whole temperature range, see Figure 5, the right panel. Only at the critical value $\kappa = \kappa_c = 1$, the density $n^{(+)}(T)$ touches the critical curve $n_{\lim}(T)$ at the temperature $T = T_*$. For this special case where $\kappa = 1$, we have calculated the heat capacity and its derivative, see Figure 7. One can see that heat capacity (left panel) has a pronounced peak at a relatively high temperature of $\sim$190 MeV.

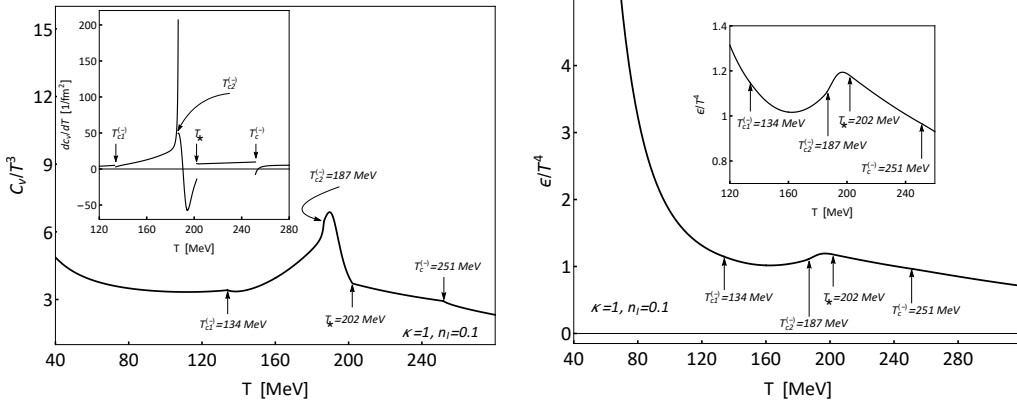

**Figure 7. Left panel:** Heat capacity normalized by $T^3$ as a function of temperature in the interacting particle–antiparticle system at $\kappa = 1$ (black solid curve). The isospin (charge) density is kept constant, $n_I = 0.1$ fm$^{-3}$. The derivative of heat capacity is shown in a small window. **Right panel:** Energy density normalized by $T^4$ versus temperature for the same system and conditions as in the left panel (black solid curve). The enlarged central area of the graphic is shown in a small window.

It is necessary to point out that the heat capacity and energy density are the physical quantities, which reflect the integrated behavior of the total particle–antiparticle system. That is why the curves $c_v(T)$ and $\varepsilon(T)$ "carry" specific peculiarities that are due to the joined behavior of the particles and antiparticles. This can be seen clearly in Figure 7 for $\kappa = 1$. Indeed, we see three phase transitions of second order at $T = T_{c1}$, $T_{c2}$, $T_c$ that are due to behavior of $\pi^-$ mesons at $\kappa = 1$. Meanwhile, for the $\pi^+$ meson subsystem at $\kappa = 1$, one can see the virtual second-order phase transition at $T = T_*$, marked as the filled star on the critical curve in Figure 5, right panel. It is a specific phase transition of the second order because there is no creation of the condensate in both directions from the temperature $T_*$ [5]. The character of this phase transition is clearly seen in Figure 7, in the small window in left panel as a discontinuity of the heat-capacity derivative at $T = T_*$. At the same time, we see a smooth behavior of the energy density at this temperature, see the small window in Figure 7 in the right panel.

We notice that all crosses of the dependencies $n^{(-)}(T)$ and $n^{(+)}(T)$ with the critical curve $n_{\text{lim}}(T)$ are exhibited as the finite discontinuity of the derivatives of heat capacity $c_v(T)$ at the temperatures $T = T_{c1}, T_{c2}, T_*, T_c$, see the left panel in Figure 7. In the right panel in this figure, we plot the energy density. One can recognize that it is really the second-order phase transitions at these four temperature points because the dependence of the energy density, $\varepsilon(T)$, is indeed continuous and without release of the latent heat.

Therefore, regarding thermodynamic behavior of the particle–antiparticle bosonic system at "weak" attraction ($\kappa \leq 1$), we identified the phase transitions of the second order at every cross point of the density $n^{(-)}(T)$ with the critical curve $n_{\text{lim}}(T)$ defined in Equation (63). For parameter $\kappa$ in the interval $0 \leq \kappa < \kappa_s$, we fix the onset of condensation at one temperature $T = T_c^{(-)}$, corresponding to a phase transition of the second order. However, for the values of parameter $\kappa$ in the interval $\kappa_s < \kappa \leq 1$, we find the onset of condensation at three temperatures $T_{c1}^{(-)}$, $T_{c2}^{(-)}$ and $T_c$ due to an oscillating behavior of the curve $n^{(-)}(T)$ around the line $n_{\text{lim}}(T)$.

The density dependence $n^{(+)}(T)$ of $\pi^+$ mesons at $\kappa = 1$ provides a remarkable feature that we once noted above. As one can see in Figure 5, in the right panel, at the temperature $T_* = 202$ MeV, the curve $n^{(+)}(T)$, calculated at $\kappa = 1$, touches the critical curve $n_{\text{lim}}(T)$, but it does not cross it. Let us look at this in some detail. For the value $\kappa = 1$, the roots (54) of equation $U(n) + m = 0$ coincide with one another: $n_1 = n_2 \equiv n_*$, where $n_* = \sqrt{m/B}$. At this density, because $U(n_*) + m = 0$, the condition (62) of the condensate creation leads to zero value of the chemical potential, i.e., $\mu_I = 0$, but $U(n_*) = -m$. Therefore, for the particle-density point $n = n_*$, the arguments in the Bose–Einstein distribution functions of the densities $n^{(+)}$ and $n_{\text{lim}}$ coincide and equal to $(\omega_k - m)/T$. Hence, it is possible to calculate the temperature that corresponds to the total particle density $n_*$ by solving the following equation: $n_{\text{lim}}(T_*) = (n_* - n_I)/2$. One can see the behavior of the chemical potential vs. temperature at $\kappa = 1$ in Figure 8 in the left panel as the blue solid curve (the axis indicating the value of the chemical potential is on the right of the graph). The chemical potential drops down to zero at one point $T = T_*$, where the density of $\pi^+$ mesons touches the critical curve, see Figure 5, the right panel. As can be seen in Figure 7, in the left panel, the common point of the line $n^{(+)}(T)$ with the line $n_{\text{lim}}(T)$ is sufficient to create a finite discontinuity of the derivative of the heat capacity with a continuous behavior of the energy density, that is, to cause a phase transition of the second order at the temperature $T_*$. We name this phenomenon the virtual phase transition of the second order because it does not lead to the creation of the condensate that plays a role of the order parameter.

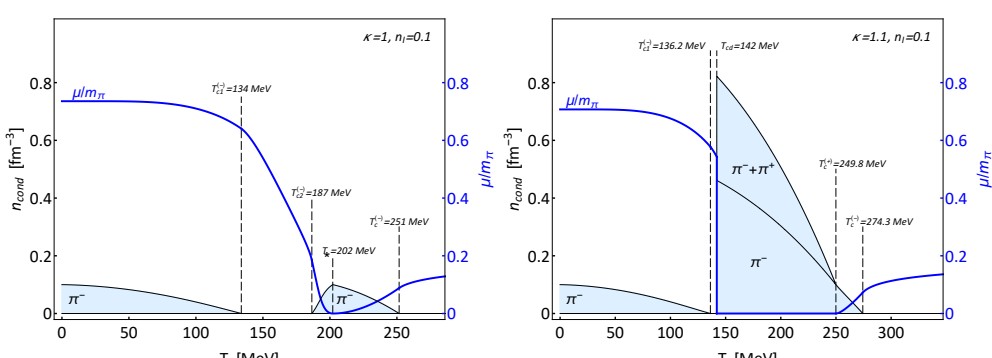

**Figure 8.** **Left panel:** Density of the condensate of $\pi^-$ mesons as a function of temperature in the interacting particle–antiparticle gas at $\kappa = 1$. The isospin (charge) density is kept constant, $n_I = 0.1$ fm$^{-3}$. Shaded blue areas show the condensate states of $\pi^-$ mesons. The blue solid line shows the behavior of the chemical potential. **Right panel:** The same as in the left panel but for $\kappa = 1.1$. The sail-like shaded area indicates the condensate states created by $\pi^-$ mesons and by $\pi^+$ mesons at the same time. The gap of the chemical potential at $T = T_{cd}$ reflects phase transition of the first order, which creates the condensate of $\pi^-$ and $\pi^+$ mesons.

At the end of this section, we can formulate the following theorem: *Each intersection of the particle density curve $n^{(\pm)}(T)$ with the critical curve $n_{\mathrm{lim}}(T)$ (or even touching the critical curve) leads to a phase transition of the second order at the temperature that characterizes this intersection point. At the temperature $T_*$ corresponding to the point of touching, we encounter a virtual phase transition of the second order without the formation of a condensate, that is, without the formation of an order parameter.*

### 7. Canonical Ensemble vs. Grand Canonical Ensemble: Description of the Boson Systems in the Presence of a Condensate

*7.1. Particle-Number Conservation in an Ideal Single-Component Bosonic System*

Let us assume that in the case of the conserved charge, we want to describe the boson system in the framework of the grand canonical ensemble, where the canonical variables are $(T, \mu)$. As a starting point, let us consider an isolated ideal single-component boson gas with a conserved number of particles (next, in the framework of the grand canonical ensemble, we will consider a particle–antiparticle boson system at a conserved charge density).

It turns out that even in this case, the general procedure is not so unambiguous. First of all, one should adjust the chemical potential at high temperatures $T$, where no condensate is present in the system, at a given particle density $n$, which should be treated as a mean value. In the canonical ensemble, where the free variable is the particle density $n$, the chemical potential is found from equation

$$n = g \int \frac{d^3 k}{(2\pi)^3} f_{\mathrm{BE}}(\omega_k, \mu).$$

(64)

On the other hand, it can be represented vice versa: at some given temperature $T'$ and chemical potential $\mu'$, by using Equation (64), one can calculate the mean value $\bar{n}$, which will be adopted as a conserved particle-number density in the canonical ensemble. However, further, for other temperatures than $T'$, one has to know the chemical potential that provides the same particle density $n$. Again, it is necessary to solve Equation (64) with respect to the chemical potential to obtain a dependence $\mu(T, n)$. The solution of Equation (64) is represented in Figure 9 in the left panel for two densities $n = 0.1 \, \mathrm{fm}^{-3}$ and $n = 0.2 \, \mathrm{fm}^{-3}$, where the critical curve is depicted as $n_{\mathrm{lim}}^{(\mathrm{id})}$. It should be noted that in the condensed phase $T < T_{\mathrm{c}}$, the chemical potential is equal to the maximum value, which is the mass of particles $\mu = m$. Then, in the condensate phase, the variables $(T, \mu)$ determine only the density of thermal particles in this temperature interval, see two examples of curves in Figure 9 in the right panel. In addition to this, it should be noted that if the chemical potential participates in the condition of condensate formation, i.e., $\mu = m$, then from a formal point of view, it cannot be a free variable in the condensate phase.

Therefore, if we continue to investigate the conservation of the number of particles in a single-component ideal gas over a wide temperature interval, we must use the chemical potential profile depicted in Figure 9 in the left panel. Then, indeed, if we use this function $\mu(T, n)$ in Equation (64) to calculate the particle density, the resulting dependence $n(T)$ actually remains constant, $n(T) = \bar{n} = \mathrm{const}$. However, in fact, this is not the use of the grand canonical ensemble, where the two free variables $(T, \mu)$ should determine the thermodynamic state of the system, we see that the chemical potential profile is calculated with the help of some value of $n$. This especially applies to the condensate phase, where the chemical potential is limited by the condition of condensate formation, i.e., $\mu = m$.

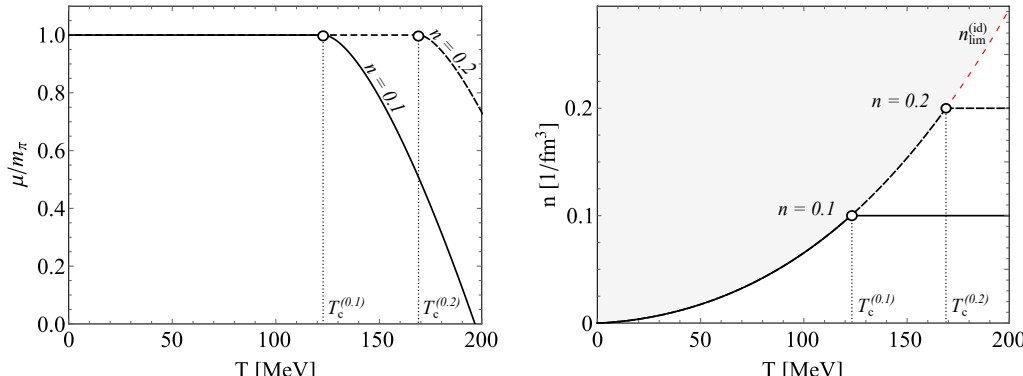

**Figure 9. Left panel:** Chemical potential vs. temperature in an ideal single-component boson gas at conserved mean value $n$ of the particle-number density for two samples: $n = 0.1$ fm$^{-3}$ with $T_c^{(0.1)}$ (the black solid line) and $n = 0.2$ fm$^{-3}$ with $T_c^{(0.2)}$ (the black dashed line). The segment $\mu = m$ belongs to the condensate phase. **Right panel:** Density of thermal particles vs. temperature in an ideal single-component boson gas. The critical curve $n_{\text{lim}}^{(\text{id})}$ is defined in (3). (The same notations as in the left panel).

The picture obtained becomes even more striking when we study the conservation of charge in a relativistic ideal boson gas of particles and antiparticles at $n_I \neq 0$. Indeed, if we assume that particles and antiparticles are simultaneously in the condensate phase, then two conditions must be satisfied simultaneously: $m - \mu_I = 0$ and $m + \mu_I = 0$, where $\mu_I$ is the isospin chemical potential, which corresponds to $n_I$. This leads to two equations: $m = 0$ and $\mu_I = 0$. As we can see, the first equation is impossible or unphysical. That is, only one condition can be fulfilled, for example $m - \mu_I = 0$. Therefore, we can formulate the following theorem: *in a relativistic bosonic ideal gas of particles and antiparticles with a conserved charge $n_I \neq 0$, only one component of the system can form a condensate phase.* The sign of the excess charge, the modulus of which is equal to $n_I$, determines the answer, which component of the system, particles or antiparticles, can be in the condensate.

### 7.2. Charge Conservation in an Interacting Particle–Antiparticle Boson System

A similar paradoxical picture arises when describing an interacting particle–antiparticle bosonic system at a finite isospin (charge) density $n_I \neq 0$ within the grand canonical ensemble. With "strong" attraction, when the temperature rises from zero, the system has a different phase structure in different temperature intervals, as was the case with "weak" attraction.

As we saw in the previous Section 6.1, with "weak" attraction, the boson system has a different phase structure in different temperature intervals. With a "strong" interaction, an additional thermodynamic state arises, when both components, that is, particles and antiparticles, can simultaneously be in the condensate phase. Therefore, if $\kappa > 1$, it is necessary to sequentially solve three sets of equations, each of which corresponds to a certain thermodynamic phase:

(a) at low temperatures, when the charge-dominant component of the particle–antiparticle system is in the condensate phase [6] and the low-density component is only in the thermal phase, this is a set of Equations (60) and (61);

(b) when both components, i.e., mesons $\pi^-$ and $\pi^+$, are in the condensate phase, it is necessary to modify set (a), see hereinafter;

(c) at high temperatures, it is a set of Equations (58) and (59), which defines the state when both components of the system, that is, particles and antiparticles, are only in the thermal phase.

There is a delicate issue when both particles and antiparticles undergo the Bose–Einstein condensation at the same time. In this case, in addition to the condensate condition (62) for $\pi^-$ mesons, the argument of the distribution function for $\pi^+$ mesons must satisfy a similar

condition to ensure that this component of the system is also present in the condensate at the same temperature $T$ and chemical potential $\mu$. Therefore, when both particles and antiparticles are in the condensate, we obtain two conditions simultaneously:

$$U(n) - \mu_I + m = 0,\tag{65}$$

$$U(n) + \mu_I + m = 0.\tag{66}$$

Then, Equations (60) and (61) should be modified to take these conditions into account. We must include a condensate component $n_{\text{cond}}^{(+)}$ of $\pi^+$ mesons, accounting for the fact that the density of thermal $\pi^+$ mesons is now $n_{\text{lim}}(T)$, as well as the density of thermal $\pi^-$ mesons. Hence, when both components are in the condensate, the set of self-consistent equations reads (case (b))

$$n = n_{\text{cond}}^{(-)}(T) + n_{\text{cond}}^{(+)}(T) + 2\,n_{\text{lim}}(T),\tag{67}$$

$$n_I = n_{\text{cond}}^{(-)}(T) - n_{\text{cond}}^{(+)}(T).\tag{68}$$

It turns out that the solutions of sets (a) and (b) exist in the same temperature interval. Indeed, in addition to self-consistent solutions of equation (a), there are two other branches of solutions: $(n_1^{(-)} = \text{const}, n_1^{(+)} = \text{const})$ and $(n_2^{(-)} = \text{const}, n_2^{(+)} = \text{const})$, which satisfy Equations (67) and (68). It can be shown that the branch $(n_2^{(-)} = (n_2 + n_I)/2, n_2^{(+)} = (n_2 - n_I)/2$, where $n_2$ is the root (54) of equation $U(n) + m = 0$, is preferable because of the higher pressure corresponding to these states.

The competition between branches (a) and (b) is resolved in the standard way according to the Gibbs criterion: the state corresponding to the highest pressure is preferred in the thermodynamic realization. Using this rule we find the temperature $T_{\text{cd}}$ from equation $p_{(a)}(T, n_I) = p_{(b)}(T, n_I)$, where the pressure $p_{(a)}(T, n_I)$ corresponds to solutions of the set of equation (a) and $p_{(b)}(T, n_I)$ to the set of equation (b). For temperatures above $T_{\text{cd}}$, the pressure that corresponds to the states determined by set (b) dominates, i.e., $p_{(b)}(T, n_I) > p_{(a)}(T, n_I)$. This leads to the transition from branch (a) to branch (b) of self-consistent solutions, which leads to a phase transition of the first order at the temperature $T = T_{\text{cd}}$.

The set of Equations (65) and (66) can be rewritten as

$$\mu_I = 0,\tag{69}$$

$$U(n) + m = 0.\tag{70}$$

Note, in Ref. [10], the system of pions was studied in the grand canonical ensemble at $\mu_I = 0$ in the mean-field approach, and the condition for the onset of the condensate phase leads to the same equation $U(n) + m = 0$.

Results of the numerical solution of the sets of equations (a), (b) and (c) for the particle density at $\kappa = 1.1$ are shown in Figure 10 in the left panel. The density $n^{(-)}(T)$ of $\pi^-$ mesons is represented by a solid blue curve, which consists of several horizontal segments and one vertical segment, which reflects a phase transition of the first order. The density $n^{(+)}(T)$ of $\pi^+$ mesons is depicted as a dashed blue curve, which also consists of several horizontal segments and one vertical segment, which also reflects a first-order phase transition. It can be seen from the figure that the isospin (charge) density in the system of bosons under consideration remains constant. Indeed, for each temperature point on the graph, it can be seen that $n^{(-)}(T) - n^{(+)}(T) = 0.1\,\text{fm}^{-3}$.

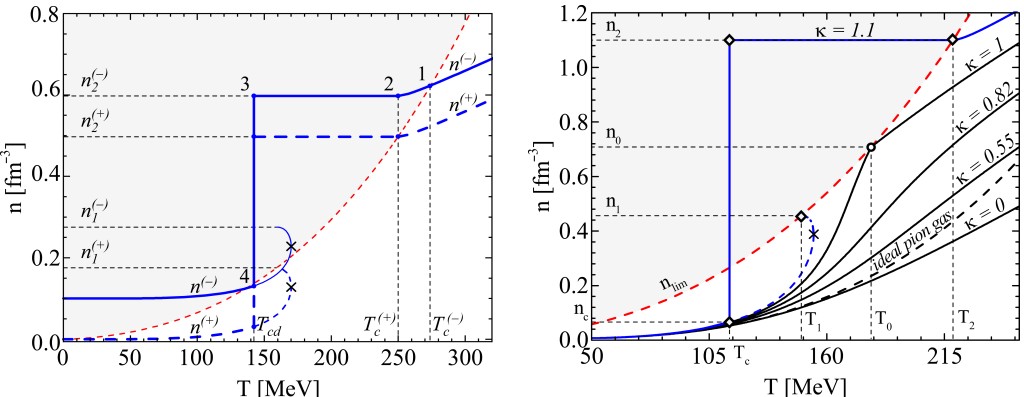

**Figure 10.** Interacting particle-antiparticle boson system in the thermodynamic mean-field model. **Left panel:** Particle densities vs. temperature at conserved isospin (charge) density $n_I = 0.1$ fm$^{-3}$ as the solid blue line consisting of several segments ($\pi^-$ mesons) and the dashed blue line consisting of several segments ($\pi^+$ mesons). The vertical segment for both dependencies indicates a phase transition of the first order with the creation of the condensate. In the condensate phase, $\mu_I = 0$. A dashed red line is the critical curve $n_{\lim}(T)$, see Equation (63). **Right panel:** Particle-number densities vs. temperature at $n_I = 0$ (or at $\mu_I = 0$): (1) the supercritical attraction $\kappa = 1.1$ is shown as a solid blue line consisting of several segments; the vertical segment (solid blue line) indicates a phase transition of the first order with the creation of the condensate; (2) particle densities at "weak" attraction $\kappa \leq 1$ are shown as solid black lines in the thermal phase. A dashed red line is the critical curve. Crosses on both panels separate metastable and non-physical states.

For a clearer comparison in the right panel in Figure 10, we took the liberty of depicting the right panel of Figure 3 once again. We would like to emphasize that in the condensate phase, both systems are represented by a zero chemical potential regardless of whether the particle–antiparticle system described in the left panel has a finite charge density, i.e., $n_I = 0.1$ fm$^{-3}$, while the particle–antiparticle system described in the right panel is characterized by zero charge density, i.e., $n_I = 0$. Therefore, if one intends to study both systems, one system with a finite charge density and another with zero charge density within the grand canonical ensemble, then the canonical variables should be $(T, \mu_I = 0)$ when describing the condensate phase in both systems.

We seem to be coming to a kind of contradiction, since the textbooks say that the chemical potential should reflect charge conservation or particle-number density conservation, as we saw in the previous Section 7.1. The resolution of this contradiction occurs according to the statement that *the grand canonical ensemble with canonical variables $(T, \mu)$ is suitable for describing only the thermal phase or for describing particles that are in kinetic states, but not in condensed states*. We verified that this is true for particle-number conservation in the case of an ideal gas of bosons, where with the canonical variables $(T, \mu)$ in the condensate phase we were able to describe only the kinetic particles, see Section 7.1.

This is also the case in our particular consideration of the relativistic particle–antiparticle boson system with conserved isospin charge $n_I$. Indeed, it can be seen in Figure 10 in the left panel in the temperature interval that corresponds to the condensate phase, i.e., between points 2 and 3 on the graph, that for each temperature from this interval, the thermal density of $\pi^-$ mesons is equal to the thermal density of $\pi^+$ mesons, since these two densities are equal to $n_{\lim}(T)$. In other words, these kinetic densities are equal to the critical curve density. Therefore, the charge density, which is determined only by thermal particles and antiparticles, is zero. Respectively, the chemical potential, which corresponds to the charge of the system, which is determined only by thermal particles, is also zero. In addition, we see that the chemical potential $\mu_I$ is really useful for describing only thermal or kinetic particles. Actually, this can be understood from the very beginning, because the chemical potential "works" in the integral (in the distribution function), which determines only the density of kinetic particles.

The results of solving the self-consistent equations for $\kappa = 1.1$ are shown in Figure 8 in the right panel (the axis indicating the value of the chemical potential is on the right side of the graph). One can see the behavior of the chemical potential, the value of which is actually zero in the phase where the particles and antiparticles are in a condensed state. It can be seen that the chemical potential drops to zero at the temperature $T_{\mathrm{cd}}$, which indicates a phase transition of the first order. We can also see in Figure 8 in the right panel that condensate forms in two different temperature intervals, at low temperatures the presence of condensate is exclusively due to charge conservation, but at higher temperatures, the formation of condensate is caused by supercritical attraction between particles.

### 7.3. Other Examples

Consider the thermodynamic mean-field model, where the mean field also depends on the isospin density. As shown in Ref. [35], since $n$ and $n_I$ are independent thermodynamic variables, the form of this mean field is as follows: $U^{(\mp)}(n, n_I) = U(n) \mp U_I(n_I)$, where $U_I(n_I)$ is an odd function, for example, $U_I(n_I) \propto n_I$, and the field $U^{(-)}$ acts on $\pi^-$ mesons, while $U^{(+)}$ acts on $\pi^+$ mesons. Then, if $\pi^-$ and $\pi^+$ mesons are in the condensate phase, two necessary conditions must be fulfilled: $m + U(n) - U_I(n_I) - \mu_I = 0$ and $m + U(n) + U_I(n_I) + \mu_I = 0$. From here, we obtain the equivalent equations: $m + U(n) = 0$ and $\mu_I = -U_I(n_I)$. Therefore, the chemical potential is fixed by the condition of condensate formation and is determined by the isospin density, which remains constant. Hence, when the mean interaction in the system depends on the isospin (charge) density, we again conclude that $\mu_I$ cannot be a free variable in the presence of a condensate, and hence, the grand canonical ensemble is not applicable in the condensate phase.

When describing the interacting particle–antiparticle bosonic system at a finite isospin (charge) density $n_I \neq 0$ in the field-theoretic approach formulated in Section 3, we encounter exactly the same paradox. Indeed, for development of the condensate by both particles and antiparticles, two conditions must be met: $M^2 - \mu_I = 0$ and $M^2 + \mu_I = 0$, where $M$ is the effective mass of quasi-particles. By complete analogy with case (c) discussed above, these conditions lead to two equations: $M^2 = 0$ and $\mu_I = 0$. Therefore, it turns out that the system with a finite charge density $n_I \neq 0$ is characterized by zero value of the chemical potential. On the other hand, we see that in the presence of condensate, the density of thermal particles is the same in the negatively and positively charged components of the system, i.e., $n_{\mathrm{th}}^{(-)}(T) = n_{\mathrm{th}}^{(+)}(T)$. Hence, the problem can be resolved by accepting that the chemical potential is responsible only for thermal (kinetic) particles.

## 8. Conclusions

Therefore, in the present study, we have investigated the relativistic interacting system of Bose particles and antiparticles, which we conventionally named "pions" due to zero spin and mass $m = 139$ MeV/c$^2$. The repulsion between particles was fixed (hard-core repulsion), but attraction between particles, which was parameterized by the dimensionless parameter $\kappa$, changes from zero ($\kappa = 0$) to some supercritical value ($\kappa > 1$).

We proved, and by this we confirmed the conclusion obtained in [8], that at "weak" attraction ($\kappa \leq 1$), the $\pi^-$ component of the system only can develop the Bose–Einstein condensate, the $\pi^+$ component is in the thermal phase for all temperatures. We have shown that for $0.93 \leq \kappa \leq 1$, in addition to the condensate of $\pi^-$ mesons at low temperatures, it can appear again in some interval at higher temperatures.

- The intersections of the particle density curves with the critical curve indicate second-order phase transitions in the system.
- At the point where the particle density of $\pi^-$ mesons touches the critical curve, the virtual phase transition of second order, i.e., a phase transition without setting the order parameter, appears.
- The meson system develops a first-order phase transition for sufficiently strong attractive interactions via forming a Bose condensate, thus releasing the latent heat. The

model predicts that the condensed phase is characterized by a constant total density of particles.

- The grand canonical ensemble cannot describe the state of the condensate since the chemical potential $\mu_I$ is significantly affected by the conditions of condensate formation, so it cannot be used as a free variable if the system is in the condensed phase. That is why the grand canonical ensemble is not suitable for describing a multi-component system in the condensate phase, even if only one of the components is in the condensate.

**Author Contributions:** writing—original draft preparation and editing, D.A.; writing—review and editing, V.G., D.Z., V.K., I.M. and H.S. All authors have read and agreed to the published version of the manuscript.

**Funding:** This research received no external funding.

**Data Availability Statement:** Not applicable.

**Acknowledgments:** D.A. is very grateful to J. Steinheimer and O. Philipsen for useful discussions and comments and greatly appreciates the warm hospitality and support provided by FIAS administration and the scientific community. The work of D.Zh. and D.A. was supported by the Simons Foundation and by the Program "The structure and dynamics of statistical and quantum-field systems" of the Department of Physics and Astronomy of the NAS of Ukraine. I.M. thanks FIAS for the support and hospitality. H.St. thanks for the support from the J. M. Eisenberg Professor Laureatus of the Fachbereich Physik.

**Conflicts of Interest:** The authors declare no conflict of interest.

### Abbreviations

The following abbreviations are used in this manuscript:

SMF   Scalar mean-field model
TMF   Thermodynamic mean-field model
FED   Free energy density

### Appendix A. Thermodynamically Consistent Mean-Field Model for the Interacting Particle–Antiparticle System

The consideration in this section is based on the thermodynamic mean-field model developed in Ref. [33], where a multi-component system consisting of any number of species was studied. Here, we consider specific equations of the thermodynamic mean-field model for the system of particles and antiparticles.

We limit our study to the case where at a fixed temperature, the interacting boson particles and boson antiparticles are in dynamical equilibrium with respect to annihilation and pair-creation processes. To take into account the interaction between the bosons, we introduce a phenomenological Skyrme-like mean field $U(n)$, which depends only on the total density of mesons $n$.

To start with, let us consider a thermodynamic system consisting of two sorts of particles. The free energy of the system and its differential can be written as

$$F(N_1, N_2, T, V) = \mu_1 N_1 + \mu_2 N_2 - pV, \tag{A1}$$

$$dF = \mu_1 dN_1 + \mu_2 dN_2 - SdT - pdV, \tag{A2}$$

where $N_{1,2}$ is the number of particles of the first and second sorts, $\mu_{1,2}$ are their chemical potentials, $p$ is the pressure in the system and $S$ and $V$ are its entropy and volume. The differential of the free energy density (FED), which, for a homogeneous system, is defined as $\Phi = F/V$, reads

$$d\Phi(n_1, n_2, T) = \mu_1 dn_1 + \mu_2 dn_2 - sdT, \tag{A3}$$

where $s = S/V$, $n_{1,2} = N_{1,2}/V$ are the entropy density and the particle number density, respectively. The chemical potentials are expressed as

$$\mu_1 = \left(\frac{\partial \Phi}{\partial n_1}\right)_T, \tag{A4}$$

$$\mu_2 = \left(\frac{\partial \Phi}{\partial n_2}\right)_T. \tag{A5}$$

We assume that the FED of a system of interacting particles can be represented as a sum of FEDs of the system without interaction $\Phi_1^{(0)} + \Phi_2^{(0)}$ and the term $\Phi_{\text{int}}$ responsible for interaction, which, in turn, depends on the total density of particles $n = n_1 + n_2$,

$$\Phi(n_1, n_2, T) = \Phi_1^{(0)}(n_1, T) + \Phi_2^{(0)}(n_2, T) + \Phi_{\text{int}}(n_1 + n_2, T). \tag{A6}$$

Then, in accordance with Equations (A4) and (A5), we obtain

$$\mu_1 = \frac{\partial \Phi_1^{(0)}}{\partial n_1} + \frac{\partial \Phi_{\text{int}}}{\partial n} = \mu_1^{(0)} + \frac{\partial \Phi_{\text{int}}}{\partial n}, \tag{A7}$$

$$\mu_2 = \frac{\partial \Phi_2^{(0)}}{\partial n_2} + \frac{\partial \Phi_{\text{int}}}{\partial n} = \mu_2^{(0)} + \frac{\partial \Phi_{\text{int}}}{\partial n}. \tag{A8}$$

The pressure can be written as

$$\begin{aligned} p(n_1, n_2, T) &= \mu_1 n_1 + \mu_2 n_2 - \Phi(n_1, n_2, T) \\ &= \left\{\mu_1^{(0)} n_1 - \Phi_1^{(0)}\right\} + \left\{\mu_2^{(0)} n_2 - \Phi_2^{(0)}\right\} + \left\{n\frac{\partial \Phi_{\text{int}}}{\partial n} - \Phi_{\text{int}}\right\}. \end{aligned} \tag{A9}$$

We introduce the following notations

$$U(n, T) = \left[\frac{\partial \Phi_{\text{int}}(n, T)}{\partial n}\right]_T, \tag{A10}$$

$$P(n, T) = n\left[\frac{\partial \Phi_{\text{int}}(n, T)}{\partial n}\right]_T - \Phi_{\text{int}}(n, T). \tag{A11}$$

From these definitions, one immediately obtains a relation that connects these two quantities

$$n\frac{\partial U(n, T)}{\partial n} = \frac{\partial P(n, T)}{\partial n}. \tag{A12}$$

Next, in Equation (A9), we use expressions for the pressure in the single-particle ideal gas

$$p_1^{(0)} = \mu_1^{(0)} n_1 - \Phi_1^{(0)} = \frac{g}{3} \int \frac{d^3 k}{(2\pi)^3} \frac{k^2}{\omega_k} f(\omega_k; \mu_1^{(0)}), \tag{A13}$$

$$p_2^{(0)} = \mu_2^{(0)} n_2 - \Phi_2^{(0)} = \frac{g}{3} \int \frac{d^3 k}{(2\pi)^3} \frac{k^2}{\omega_k} f(\omega_k; \mu_2^{0)}), \tag{A14}$$

where $f(\omega_k; T, \mu^{(0)})$ is the Bose–Einstein distribution function of ideal gas

$$f\left(\omega_k; \mu^{(0)}\right) = \left\{\exp\left[\frac{\omega_k - \mu^{(0)}}{T}\right] - 1\right\}^{-1} \quad \text{with} \quad \omega_k = \sqrt{m^2 + k^2}. \tag{A15}$$

Using relations (A7) and (A8) in the form

$$\mu_1^{(0)} = \mu_1 - U(n), \tag{A16}$$

$$\mu_2^{(0)} = \mu_2 - U(n), \tag{A17}$$

one can insert these expressions into Equations (A13) and (A14) and then rewrite the total pressure (A9) as

$$p(T, n_1, n_2) = \frac{g}{3} \int \frac{d^3k}{(2\pi)^3} \frac{k^2}{\omega_k} \left[ f(E(\boldsymbol{k}, n); \mu_1) + f(E(\boldsymbol{k}, n); \mu_2) \right] + P(T, n), \tag{A18}$$

where $g$ is the degeneracy factor, $E(\boldsymbol{k}, n) = \sqrt{m^2 + \boldsymbol{k}^2} + U(T, n)$ is the effective single-particle energy and $P(T, n)$ can be treated now as the excess pressure.

To obtain a self-consistent equation for the total particle-number density $n$, it is convenient to pass from the variables $(T, n_1, n_2)$ to the variables $(T, \mu_1, \mu_2)$. In this case, the total number of particles $n$ in the system is also a function of new variables $(T, \mu_1, \mu_2)$. Then, for the total number of particles $n$, we obtain

$$
\begin{aligned}
n &= n_1 + n_2 = \left(\frac{\partial p}{\partial \mu_1}\right)_T + \left(\frac{\partial p}{\partial \mu_2}\right)_T \\
&= g \int \frac{d^3k}{(2\pi)^3} \left[ f(E(\boldsymbol{k}, n); \mu_1) + f(E(\boldsymbol{k}, n); \mu_2) \right].
\end{aligned} \tag{A19}
$$

For free energy density, one has (see Equation (A1)) an expression

$$\Phi = \mu_1 n_1 + \mu_2 n_2 - p. \tag{A20}$$

*The System of Particles and Antiparticles*

The chemical potential $\mu$ includes components related to different quantum numbers

$$\mu = B\mu_B + S\mu_S + Q\mu_Q + I\mu_I + \dots, \tag{A21}$$

where $B$, $S$, $Q$ and $I$ correspond to the baryon quantum number, strangeness, electric charge and isospin, respectively. It is clear that the chemical potentials of boson particles $\mu_1$ and boson antiparticles $\mu_2$ have opposite signs [33]

$$\mu_1 = -\mu_2 \equiv \mu_I. \tag{A22}$$

Requiring the conservation of the isotopic spin $n_I$ in the system, we obtain the set of equations

$$n = g \int \frac{d^3k}{(2\pi)^3} \left[ f(E(\boldsymbol{k}, n); \mu_I) + f(E(\boldsymbol{k}, n); -\mu_I) \right], \tag{A23}$$

$$n_I = g \int \frac{d^3k}{(2\pi)^3} \left[ f(E(\boldsymbol{k}, n); \mu_I) - f(E(\boldsymbol{k}, n); -\mu_I) \right], \tag{A24}$$

where the Bose–Einstein distribution function reads

$$f(E; \mu) = \left\{ \exp\left[\frac{E - \mu}{T}\right] - 1 \right\}^{-1}. \tag{A25}$$

The set of Equations (A23) and (A24) can be solved with respect to the thermodynamic quantities $n$ and $\mu_I$ for given canonical variables $T$ and $n_I$. As a result, we obtain the functions

$$n = n(T, n_I), \quad \mu_I = \mu_I(T, n_I). \tag{A26}$$

In our case, the interaction between particles is described by the Skyrme-like mean field

$$U(n) = -An + Bn^2, \tag{A27}$$

where $n$ is the total particle-number density. Using the self-consistent solutions $n(T, n_I)$ and $\mu_I(T, n_I)$ of the set of Equations (A23) and (A24), one can obtain the expressions for the pressure and free energy density in the boson interacting system in the following form

$$p = \frac{g}{3} \int \frac{d^3k}{(2\pi)^3} \frac{k^2}{\omega_k} \left[ f(E(\mathbf{k}, n); \mu_I) + f(E(\mathbf{k}, n); -\mu_I) \right] + P(n), \tag{A28}$$

$$\Phi = n_I \mu_I(T, n_I) - p(T, n_I), \tag{A29}$$

where $E(\mathbf{k}, n) = \sqrt{m^2 + k^2} + U(n)$. [7] Here, the excess pressure $P(n)$ is known, and it can be calculated with the help of integration of relation (A12) using the given mean field (A27) and a natural initial condition $P(n = 0) = 0$. Hence, after integration, one obtains

$$P(n) = -\frac{A}{2} n^2 + \frac{2B}{3} n^3. \tag{A30}$$

With the help of free energy density, it is easy to calculate the volumetric heat capacity $c_V$

$$c_V = -T \frac{\partial^2 \Phi}{\partial T^2}. \tag{A31}$$

## Notes

[1]   In the nonrelativistic case, where $\mu_{\text{nonrel}} = \mu - m$, the maximum value of the thermal-particle density is achieved at zero chemical potential.

[2]   It should be noted that we just conventionally say "condensate phase". In fact, it is the thermodynamic state of a system that contains thermal particles and condensed particles at the same time.

[3]   In fact, the name "condensate phase" is just a conventional one because this phase is a mixture of the thermal (kinetic) particles and the condensed particles.

[4]    For our choice of the total charge of the system, it is the $\pi^-$ mesons.

[5]   Because we named it as the virtual phase transition of the second order.

[6]   For our choice of the total charge of the system, these are $\pi^-$ mesons.

[7]   It should be noted that Equations (A23), (A24) and (A28) take place only in the absence of condensate in the system.

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
