# Peer review of "Phase Transitions in the Interacting Relativistic Boson Systems"

_universe, doi:10.3390/universe9090411_

Round 1

Reviewer 1 Report

This is a very interesting paper about the Bose-Einstein condensation (BEC) for relativistic self-interacting scalar fields. The paper contains many new results, in particular about the order of the phase transition depending on the strength of the (repulsive or attractive) interaction.

In the introduction, the discussion of the previously existing literature can be improved. Some relevant historical papers about relativistic BEC are not cited. For instance,

- J. Bernstein and S. Dodelson Phys. Rev. Lett. 66, 683 (1991).

- D.J. Toms, Phys. Rev. Lett. 69, 1152 (1992).

- D.J. Toms, Phys. Rev. D 50, 6457 (1994).

- K. Shiokowa and B.L. Hu, Phys. Rev. D 60, 105016 (1999).

- L. Salasnich, Nuovo Cimento B 117, 637 (2002). 

- V.V. Begun and M.I. Gorenstein, Phys. Rev. C 73,054904 (2006).

- V.V. Begun and M.I. Gorenstein, Phys. Rev. C 77, 064903 (2008).

- G. Marko, U. Reinosa, and Z. Szep, Phys. Rev. D 90, 125021 (2014).

I strongly suggest to add some phrases in the Introduction about these previous results, discussing also how they are connected with the findings of the present paper. 

Author Response

New references and relevant text have been added to the introduction (see list of corrections).

Reviewer 2 Report

The study of an self-interacting system of relativistic bosons is an interesting subject in connection with the investigation of effective models to describe the properties of strongly-interacting matter. The authors present in their work an extension of their previously published models that considers now both particles and antiparticles. This is of particular relevance in systems at finite temperature and net densities close to zero. Although the paper includes some interesting results, which deserve to be published, I cannot recommend the publication of the manuscript in the present form and hence suggest a substantial revision. In the following I list the major problems but I shall not go into all details as it does not make sense at the present stage:

1. The manuscript is much too long, it is an exhausting reading, and the essential results are hidden among a lot of unimportant discussion. The length of the paper can be much reduced. Most of the formalism was published before in references [26-28] and does not need to be repeated here. Only the defining equations of the models (e.g. Lagrangian densities or thermodynamic potentials) and the resulting expressions for the thermodynamic quantities need to be given explicitly. Since all thermodynamic quantities are explicitly known, it is easy to give, e.g., the free energy (density) as a functions of the densities and the temperature.

2. The main theoretical parts (sections 2. - 4.) can be presented in a much more concise and clearly arranged form. It is not necessary to start with the model of noninteracting bosons, then add the interaction and finally consider also antiparticles. Just give for the two models the full defining forms and then mention the special cases.

3. The main difference of the two models, called scalar mean-field (SMF) model and thermodynamic mean-field (TMF) model, is the choice of the (self-)interaction of the fields. In the SMF model, it leads to an effective, medium dependent mass $M$; in the TMF model a potential $U$ appears in the dispersion relation $E_part(k) = \sqrt{k^2+m^2}+U$. By thermodynamic consistency reasons, I would expect that $U$ has to be replaces by $-U$ for the antiparticle because it is of "vector" type as in other relativistic mean-field models for strongly-interacting matter. It is not clear from the paper how this is considered in the TMF model.

4. The discussion of the theory and results should be arranged along the following considerations: Since there are two particle species in the general system, they should be considered individually in the theoretical formulation (as in appendix A). The fact that they are a pair of particles and antiparticles, leads to some specific conditions: 1. m_part = m_antipart, 2. M_part = M_antipart, 3. mu_part = -mu_antipart, 4. mu^eff_part = -mu^eff_antipart with the effective chemical potential mu^{eff} = mu-U (hence the minus sign in the energy E for the antiparticle). The chemical potential mu can only assume values in the interval [-E_part(k=0),E_part(k=0)] for bosons. If it is exactly at the upper (lower) border, there can be a condensate of particles (antiparticles) if the thermal part of the density is not sufficient to reach the given total (net) density n = n_part - n_antipart. If mu is inside the interval, there is no condensate. If the net density is zero, also the chemical potential is zero. A condensation of both particles and antiparticles can only appear if the above given interval shrinks to zero, i.e., M=0 (SMF model) or m+U=0 (TMF model). Then, however, it is questionable whether one wants to call it a 'condensate', cf. photons.

5. It is interesting to discuss the thermodynamic features and possible phase transitions, in particular, when the condensation appear and how the interaction affects them. A special case is given when the effective mass in the SMF model becomes zero at certain densities. Obviously, this is possible only for a sufficiently strong attractive part in the interaction.

6. I don't think that it makes sense to show the 'QGP' range in figures 7 and 10, as it is a completely different system.

7. The 'theorem' on the problem to apply the grand canonical ensemble description to the case with a condensation of particles is part of a more general observation that, e.g., appears for phase transitions. For instance, in a one-particle system with a phase transition describe with a Maxwell construction, the pressure and chemical potential are constant in the coexistence region and thus cannot used as proper variables. Nevertheless, it is possible to write down the grand canonical potential in the variables T and mu and to derive all relevant thermodynamic quantities.

Reviewer 3 Report

More emphasis on what is really novel in this  study.

Needs review of English speaker.

Author Response

1) More emphasis on what is really novel in this  study.

Section 7, page 20 Added text (lines 575-586):Section 7, page 20 Added text merged with the last paragraph of section 7 into a subsection 7.3 ”Other examples.” (see list of corrections)
